

**The susceptibility assessment of multi-hazard in the Pearl**
**River Delta Economic Zone, China**
Chuanming Ma*, Xiaoyu WU, Bin LI, Ximei Hu
*Corresponding author at: School of Environmental Studies, China University of Geosciences,
Wuhan 430074, China. Tel.: +86-27-67883159. Email: machuanming@cug.edu.cn



## Abstract

The multi-hazard susceptibility assessment can provide a basis to decision-making for land use planning and geo-hazards management. The main scope of this paper is assess multi-hazard susceptibility to identify susceptibility area by using an integrated method of the Analytic Hierarchy Process (AHP) and the Difference Method (MD) within MapGIS environment. The basic principle of this method is to predict future geological hazards based on occurrence mechanism of occurred geological hazards and the geological conditions that caused past geological hazards. Typical geo-hazards susceptibility are separately assessed by applying Analytic Hierarchy Process (AHP). The multi-hazard susceptibility is completed by synthesizing individual geo-hazards susceptibility result with the Difference Method (MD), the multi-hazard susceptibility map is generated by utilizing MapGIS platform. The multi-hazard map can provide decision-makers with visual information for geo-hazards management and land use planning, which reduce confusion of decision-makers on high number of individual geo-hazard map. The study area was categorized into high susceptibility zone, moderate susceptibility zone, low susceptibility zone, and insusceptible zone, accounting for 16.5%, 41.6%, 33.8% and 8.1% of the total study area, respectively. The multi-hazad susceptibility result can be combined with other conditions to provide decision- makers with theoretical basis for geo-hzards management and planning of development.


**Key words**: susceptibility assessment; mul-hazards; Analytic Hierarchy Process (AHP) - Difference (DM); MapGIS; The Pearl River Delta Economic Zone




## 1. Introduction


Geological hazards occur frequently, and the types of disasters in China are various
(National Disaster Mitigation Center Disaster Information Department, 2009),
especially southwest region of China (Tang and Wu, 1990). The Pearl River Delta
Economic Zone is the transitional belt and sensitive belt of geological environment,
nears the South China Sea, characterized by strong land-ocean interaction, widely
distributed Quaternary, complex geological structure, and various landform. It is
susceptible to cause geological disasters (Li, 2012). The Pearl River Delta Economic
Zone is the pilot area of China's reform and opening and an important economic
growth belt, and it plays a pivotal role in the social and economic development and
the overall situation of reform and opening, as well as a prominent leading role. 2016
annual government report of Guangdong Province states that it will launch a higher
level of development in the Pearl River Delta Economic Zone, building the
Guangdong-Hong Kong-Macao Greater Bay Area in cooperation with Hong Kong
and Macao, and ranking first among all the Bay Areas in the world. With the rapid
economic development for the Pearl River Delta Economic Zone, the strength of
development and utilization for geological environment trends to increase, the
frequency and intensity of geological hazards intensifies rapidly, which has a great
threaten upon people's lives and property (Zhang, 2012). The occurrence of
geological hazards seriously restricted the urban development and the sustainable
development of human society (Unitto and Shaw, 2016). Therefore, in order to
minimize the loss of human life and reduce economic consequences, management of
geological hazards is essential. Thus, it is very meaning to evaluate geological hazards
susceptibility and identify different susceptibility areas for prevention and
management of geological disaster.
Since geological hazards are complex phenomena, currently, various researches have
focused on a single geological hazard research (Komac, 2006; Pradhan et al., 2016;
Wang et al., 2015; Zhou et al., 2002). But, one region may suffer from more than one
geological hazard. The susceptibility assessment of multi-hazard that consists of
relative information of different hazards is important tool for geological management
and urban planning. The United Nations (UN, 2002) has emphasized the significance
of multi-hazard assessment and referred that it "is an essential element of a safer



world in the twenty-first century". However, multi-hazard susceptibility assessment is
a complex process and confronted with a challenges. At early stages, qualitative
assessment methods were widely used to evaluate geological hazards susceptibility
(Bijukchhen et al., 2013; Cui et al., 2004; Degg, 1992; Liang et al., 2011; Zhou et al.
2002), which are based on statistical analysis of the relationship between geological
hazards and different controlling factors, but it is difficult to describe the real
relationships of different influencing factors and forecast geological hazards. In recent
years, with development of science and technology, the methods that combines
qualitative and quantitative analysis are widely used to evaluate geological hazards
susceptibility (Lee et al., 2018; Wang et al., 2015; Yilmaz, 2009). One widely used
method of susceptibility assessment is the Analytic Hierarchy Process (AHP)
( Karaman, 2015; Karaman and Erden, 2014; Komac, 2006; Peng et al., 2012; Rozos
et al., 2011). The AHP is a multiple criteria decision-making that combines qualitative
and quantitative factors for ranking and evaluating alternative scenarios, among which
the best solution is ultimately chosen (Satty, 1980; Satty, 2008). Preventive measures
for different geological hazard are various, and their damage on environment and
people's lives and property is not neutralized. thus, multi-hazrd assessment is
completed by synthesizing all individual geological hazards with the Difference
Method. The principle of this method is that the geological hazards susceptibility in
this unit is considered high, as long as there is a kind of geological hazard under high
susceptibility in specific evaluation unit.
In this paper, a new method that integrated the Analytic Hierarchy Process (AHP) and
the Difference Method is proposed to assess multi-hazard susceptibility. Individual
hazard susceptibility is assessed with via of the Analytic Hierarchy Process (AHP)
and spatial analysis of MapGIS, based on the geological hazards investigation and
geological environmental conditions of the study area. The difference method is used
to assess multi-hazard susceptibility by synthesizing the five aforementioned
geohazards susceptibility assessment. Moreover, a multi-hazard susceptibility map is
produced with MapGIS. The multi-hazard susceptibility map will benefit local
governments in making policies on urban development and infrastructure layout, and
it also offer more accurate and effective theoretical and practical guide to land use
planning and site selection of major projects, coming true the maximum utilization of
limited resources and the maximum economic efficiency with limited environment.




## 2. The study area

### 2.1 Natural geographical conditions

The Pearl River Delta Economic Zone, with a total area of 41698 km$^2$, is located in the south-central Guangdong Province, China (Fig.01), nears the South China Sea, between 21°43' ~ 23°56' N latitude and 112°00' ~ 115°24' E longitude. It includes 9 prefecture-level cities.

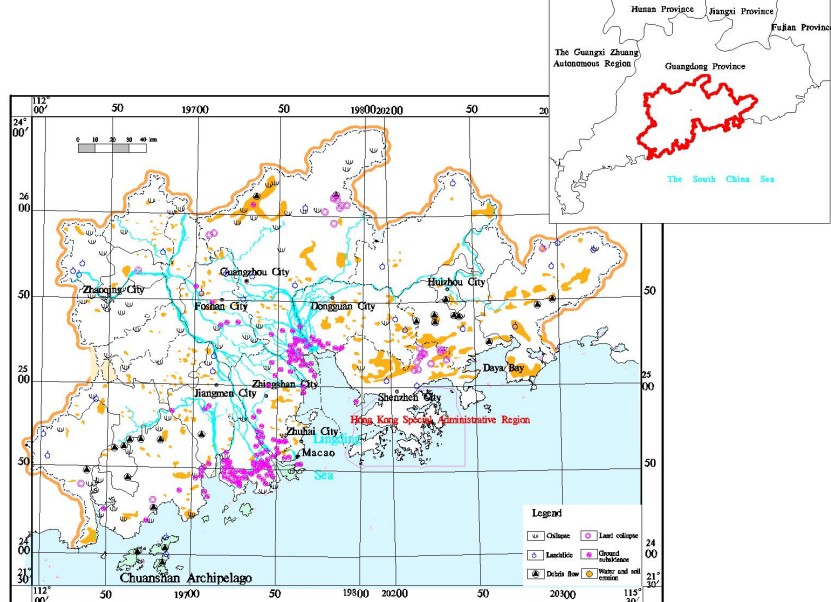

Fig.01 The map of the study area in The Pearl River Delta Economic Zone

The study area belongs to subtropical monsoon climate, characterized by mild, humid and abundant rainfall. The rainfall is characterized by large precipitation, more rainy days, stronger seasonal rainfall, and uneven spatial distribution under influence of monsoon climate. The annual precipitation is reported as about 1800-2200mm (Fig.02).

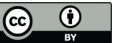


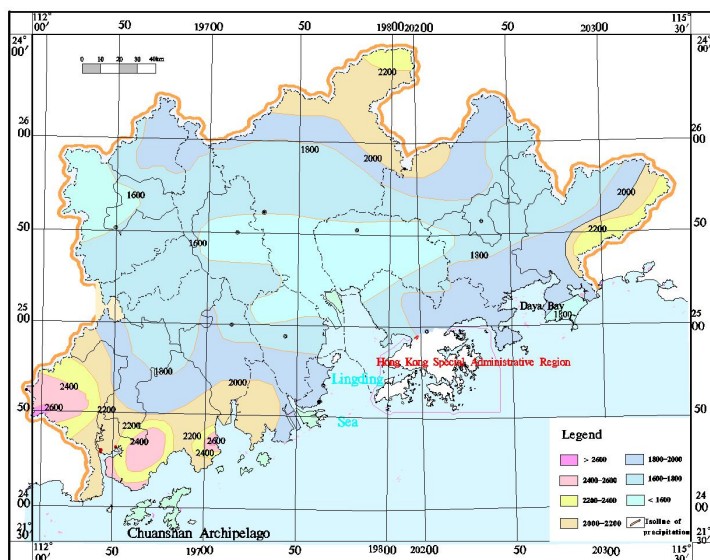

Fig.02 The precipitation map of the study area

The topography is dominated by the Pearl River delta plain, surrounded by intermittent mountain and hills, such as Gudou Mountain, Tianlu Mountain and Luofu Mountain. The terrain is smooth, ranging in altitude from -0.2 m to 0.9 m in the plain area. Based on the different genetic type, the geomorphic units are divided into 12 kinds of level Ⅱ geomorphological units, consisting of erosion and denudation middle mountains, erosion and denudation low mountains, erosion and denudation hills, erosion and denudation platforms, karst hills, volcanic hills, delta plain, alluvial and marine deposition plain, alluvial plain, alluvial and dilluvial plain, marine deposition plain and islands.





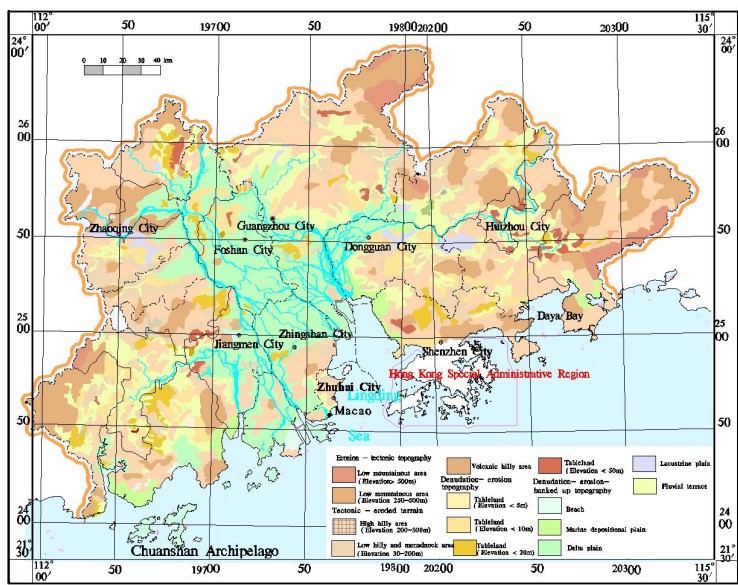

Fig.03 The topography map of the study area

## 2.2 Geological conditions

Development of the strata is relatively complete, and it is characterized by
complicated types and the wide distribution. The stratigraphic age of the outcropped
bedrock ranges from the oldest Metamorphic rocks to the latest Quaternary loose
debris deposition rocks, the outcropped strata is mainly Quaternary, followed by the
Sinian, Cambrian, Devonian, Carboniferous, Jurassic and Cretaceous. The distribution
for Mesoproterozoic, Ordovician, Permian and Paleogene are sporadic. The
outcropped Quaternary loose area accounts for 3/4 of the strata area, the outcropped
bedrock area accounts for 1/4 of the strata area. The area that develop Magmatic rocks
accounts for about 30% of the entire study area, dominated by intrusive rocks, and
volcanic rocks only develop in small areas.



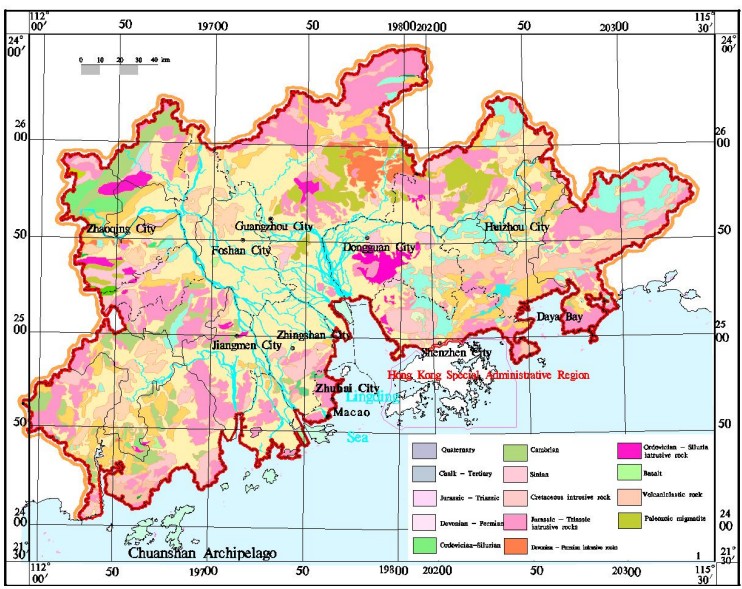

Fig.04 The geological map of the study area

## 2.3 Hydrogeological conditions

In the study area, groundwater is divided into three types: loose rock pore water,
carbonate karst water and bedrock fissure water, hydrogeological characteristics are
shown in Fig.05.

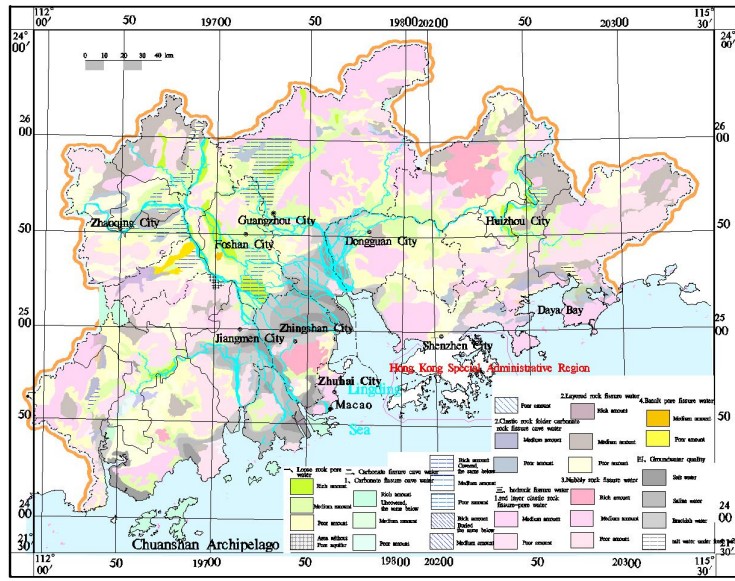




Fig.05 The hydrogeological map of the study area

## 2.4 Engineering geological condition

The rock-soil body is restricted by the topography, stratum, lithology, geological
structure, and it is also affected by the hydrogeological conditions, natural geological
conditions within the study area. Based on the nature, origin and structural features of
the rock-soil body, the rock-soil body is divided into three types: magmatic rocks,
metamorphic rocks and sedimentary rocks. In addition, it can be also divided into
gravel soil group, sandy soil group, clay soil group and intrusive rock residual soil
group, extrusive rock residual soil group and metamorphic rock residual soil group
(Fig.06).

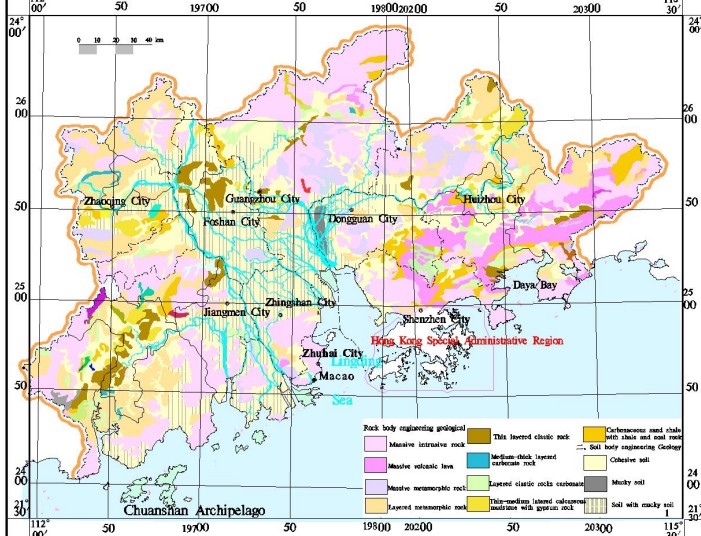

Fig.06 The engineering geological map of the study area

## 2.5 The major geological hazards

According to a field geological survey, typical geological hazards that occurred within
the study area mainly consist of collapse, landslide, ground subsidence, karst collapse,
water and soil loss, and seawater intrusion. As of 2014, there are 52 large-scale
collapses, 35 landslides and 5 debris flow have been found in the study area. In
addition to, 129 ground subsidence hazards occurred in the study area, among of them,





there are 76 ground subsidence with less than 10 cm of accumulative subsidence are
found within the study area. Water and soil erosion is fragmented distributed in
mountainous areas, hilly areas and tableland areas, which are characterized as karst
desertification, granite and less vegetation. In addition, it is widely distributed in
Longgang District, Shenzhen City and Huadu District, Guangzhou City. According to
statistics, water and soil erosion covers an area of 2300km$^2$, accounting for about
4.8% of the total land area. The seawater intrusion mainly occurred in the Pearl River
Estuary area. The scope of the annual seawater invasion spread to Yaxi Town -
Hualong Town - Humen town area. It spread to the inland area, and it possibly
reached Guangzhou City during the drought years. According to the research (Liu
2004), the driving forces of seawater intrusion for the study area are mainly tides and
runoff, followed by saltwater tides. The distribution for geological hazards is shown
in Fig.1.

## 2.6 Human activity characteristics

Except for the Pearl River Delta plain located in the hinterland, other lower-lying hills
or platforms can be reclaimed into dry land that is suitable for planting various crops,
fruit trees and economic trees. In recent years, with the rapid economic development,
the land-use structure has changed significantly. The area of cultivated land and
garden plot are declining year by year, and the construction land rapidly expand. In
the background of rapid economic and social development, the land use structure still
will has a great change in the future, and "the expansion of land for urban
construction, the massive loss of cultivated land and garden plot" will are the main
features.





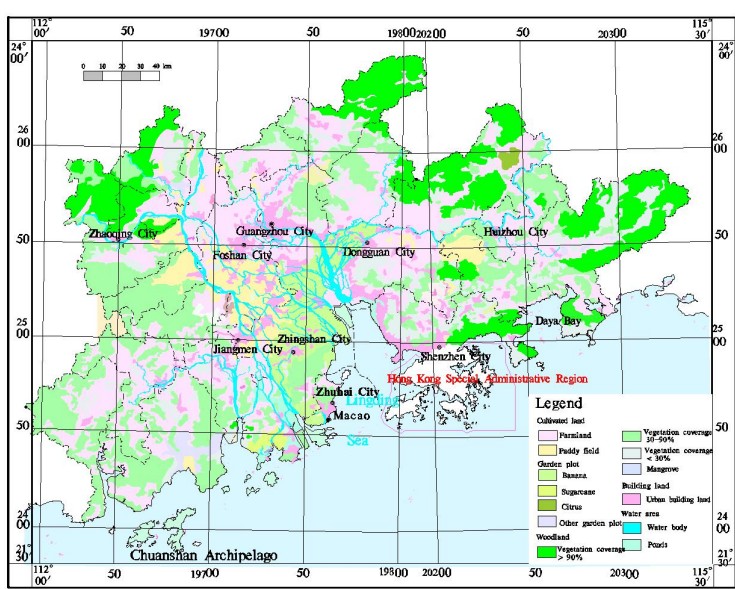


Fig.7 The land use map of the study area

## 3. Materials and methods

### 3.1 Methods

**Geological hazards causal factors**

(1) Karst collapse causal factors

Obtained research results (Su, 1998; Wang, 2001) show that the formation of karst collapse is mainly affected by degree of karst development, overburden characteristics, geological structure, and groundwater activities. Karst development is basis and prerequisite for formation of karst collapse. Overburden is material basis for formation of karst collapse and controls its formation in certain degree. Large overburden thickness can effectively disperse pressure of the soil body on the soil hole. Compared with the thinner overburden, the larger overburden thickness is less prone to karst collapse, and the scale and form of karst collapse also are closely connected with the overburden thickness. Groundwater activities is the main power producer to cause karst collapse. Geological structure can control the development of karst and can provide a good site for soil erosion, and the spatial distribution of karst development is also closely related to the geological structure. In general, the stretching direction of the karst collapse area is consistent with that of the geological structure (Fu, 2009). Based on the above analysis and geological environmental



conditions for the study area, the causal factors of karst collapse include the degree of
karst development, lithology, overburden thickness, aquifer water yield property and
the distance to the fault.
(2) Landslide and collapse causal factors
Collapses differ from landslides obviously in the form of occurrence, scale and
perniciousness, but there are also internal relations and transformation relations
between them, which make them have strong consistency in space-time distribution.
Collapses usually happen accompanied occurrence of landslides, and collapses occur
frequently in the area where landslides happen. Moreover, the causal factor of
collapses occurrence maintain basically consistency with that of landslides. Thus, this
paper carries out the susceptibility assessment of collapses and landslides.
According to the statistical analysis of geological disasters, the spatial distribution
characteristics of collapse are affected by topography, geological structure,
stratigraphic lithology and climatic and hydrological conditions. Moreover, there was
a positive correlation between the number of annul collapse and temporal distribution
of precipitation (Deng, 2008). Topography conditions are the prerequisites for
formation of landslide hazards (Li, 1996). Topographic differences provide
gravitational potential energy for instability movement of rock and soil body.
Geological conditions, characteristics of rock and soil body and hydrological
conditions also play key role in controlling slope instability. Based on analyzing
formation conditions and development characteristics of collapses and landslide
which occurred within the study area, main causal factors of collapses and landslides
include topography, lithology, the distance to fracture and precipitation.
(3) Ground subsidence causal factors
The mollisol is prerequisite factor for controlling the formation of ground subsidence,
so the area distributed with moilisol is considered as study range for ground
subsidence susceptibility. Geological settings are primary internal factor. The mollisol
distributes in the entire delta alluvial plain, and its thickness trends to increase from
the top to the front of the delta. The ground subsidence frequently occur in the central
and southern coastal areas of the study area, where the mollisol is characterized as
large thickness, shallow depth and new age of deposits formation. In general, the
degree of ground subsidence is closely related to the characteristics of millisol,
primarily including the age of millisol deposition, the thickness of millisol layer,
depth of millisol and the thickness of overburden. Hydrogeological conditions are



triggering factor for formation of ground subsidence. The ground subsidence mainly
occur in clay layer, and it is extremely sensitive to the change of groundwater table.
Thus, investigating the distribution characteristics of groundwater is prerequisite to
study ground subsidence. Stronger aquifer water yield property means larger
allowable exploit amount of groundwater, that is, the susceptibility of ground
subsidence is larger. According to survey result, it is found that ground subsidence
mostly occurred in the groundwater runoff area, and it distributed along the stretching
direction of fracture. According to the above analysis and geological investigation
result, we can found that the age of millisol deposition, the thickness of deposition,
aquifer water yield property and the distance to fracture are main causal factors of
ground subsidence.
(4) Water and soil erosion causal factors
Based on analyzing the occurrence mechanism and formation conditions of water and
soil erosion, the casual factors of water and soil erosion consist of topographic, soil
type, vegetation type, precipitation and the density of of river network for the study
area. Soil is the material basis for water and soil erosion to occur, it is also the object
to erosion. Water and soil erosion is mainly distributed on granite-developed soil. The
distribution of latosolic red soil is the mostly wide within the study area, accounting
for 44.8% of the land area, follow by is paddy soil, accounting for 40.20%. The parent
material of latored soil is mainly granite, the granite is characterized by thick
weathering soil, loose structure and poor soil viscosity. After destroying the original
vegetation and slope conditions, water and soil erosion was caused under the
long-term erosion and scour of rainfall. Rugged topography is the direct factor to
cause water and soil erosion, the steeper the slope is, the shorter the confluence time
is, the larger the runoff energy is, the stronger the erosion of water on the land is.
Water and soil erosion mainly occurred in hilly area for the study area. The vegetation
is critical factor for controlling the occurrence of water and soil erosion, because it
can prevent soil erosion, mainly including reduction for rainfall energy, water
retention and anti-erosion. Rainfall is the direct dynamic factor causing water and soil
erosion. The annual precipitation 1600 mm within the Delta plain area is less than that
of the surrounding hilly area, with annual precipitation of 2000-2600 mm.
(5) Seawater intrusion causal factors
Hydrodynamic conditions and hydrogeological conditions are two essential factors for
controlling the occurrence of seawater intrusion, the hydrodynamic condition means



that there is a certain head pressure between seawater and fresh water,
hydrogeological conditions is that there is a hydraulic relation between the seawater
and the land aquifer. When these two conditions all are available, seawater intrusion
trends to occur. Seawater intrusion was caused by the change of hydrodynamic
conditions of the coastal groundwater with the study area, major dynamics are tides
and runoff. Seawater intrusion only occurred in winter and spring in most of the
coastal areas of the study area, because precipitation is small, groundwater is not
recharged in time, resulting to lowing of groundwater table, in winter and spring (Sun,
2011). So over-exploitation of groundwater can aggravate seawater intrusion.
According to geological conditions and the situation of seawater intrusion.
Topography, the type of Quaternary sedimentary rock, groundwater table and
precipitation are main influencing factors of seawater intrusion.
**Application of the analytic hierarchy process**
The AHP method, pioneered by Saaty in the 1970s, is a multi-objective decision
analysis method that combines qualitative and quantitative analysis. A detailed
description of the AHP method is available in Saaty (1980). The procedure for using
this method can be summarized as follows (Saaty, 2008): (1) Structuring the decision
hierarchy, the assessment object is divided into a few structure layers, namely, the
target layer, criterion layer, and element layer. (2) Constructing a series of
pair-comparison judgment matrices between factors, and the pairwise comparison
employs an underlying nine-point recording assessment to rate the relative importance
on a one-to-one basis of each factor. (3) The consistency of pairwise comparison
matrix between factors should be measured by the consistency ratio (CR), which is
the consistency index of the matrix. And the value of the CR should be no higher than
0.1. CR can be calculated by Eq.(1):
CR=CI/RI             (1)
where RI is the mean random consistency index, which depends on the order of the
matrix given in Table 1; CI is the consistency index used to measure the deviation of
the matrix, as expressed in Eq.(2):
$CI = \dfrac{\lambda_{max} - 1}{n - 1}$           (2)
Where $\lambda_{max}$ is the largest or principal eigenvalue of the matrix and can be easily
calculated from the matrix, and n is the order of the matrix.



<Table 1>

(4) The factor weights are obtained through matrix operations, sorting operations and
a consistency check.
The susceptibility value of individual geological hazard is computed according to the
following formula Eq.(3):
$\text{SI} = \sum_{i=1}^{n} R_i W_i$                                                        (3)
Where SI denotes the susceptibility value, R and W are ratings and weights of the
caused factors, respectively, n is the number of factors.

**4. Results**
**4.1 Assessment of individual geological-hazard susceptibility**
According to above analysis, aforementioned causal factors of each geo-hazards is
considered as assessment indexes of individual geo-hazards susceptibility. And each
index is standardize to a uniform rating scale and each of them is assigned a attribute
value shown in Table 2. The weight of facor is assigned by applying AHP shown in
Table 2, the consistency ratio (CR) of all judgment matrix is less than o.1, which
indicates that the comparison matrix is consistent.

<Table 2>

Based on an established classification criteria of evaluation indexes for geological
hazards susceptibility shown in table 2, the susceptibility value was calculated by
using Eq.(3). Based on the equidistant division method, the susceptibility value is
divided into four classes: lowest, low, high, and highest. Based on classification of the
susceptibility value, and the study area is classified into four geo-hazard susceptibility
areas accordingly. The susceptibility map of individual geo-hazards is produced
within MapGIS 6.7 environment. First, the basic data of the study area are converted
to raster images of each factor using the image processing in MapGIS 6.7. Next, the
images are reclassified and assigned the corresponding value of each rank using
graphics processing. Finally, the susceptibility map is elaborated by overlying ranking
maps with the spatial analyst tool of MapGIS 6.7. The susceptibility map of
individual geo-hazard is shown in Fig. 08-Fig.12.






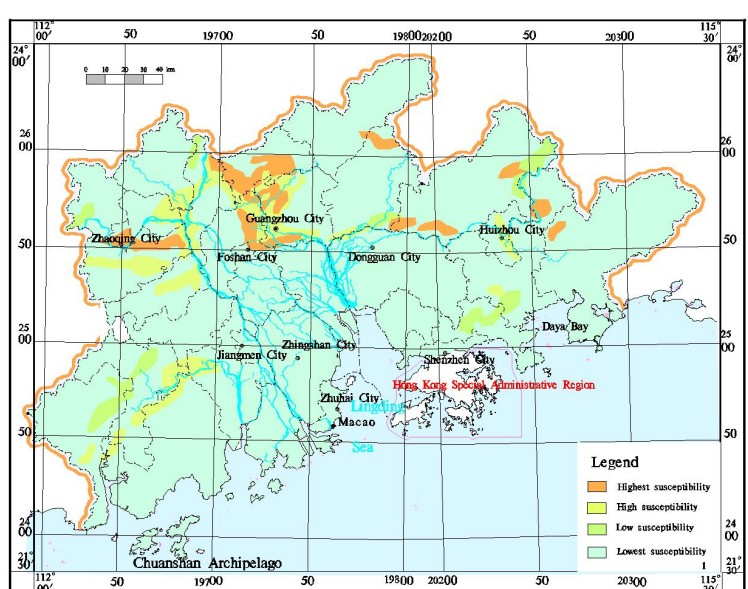


Fig.08 Karst collapse susceptibility map of the study area

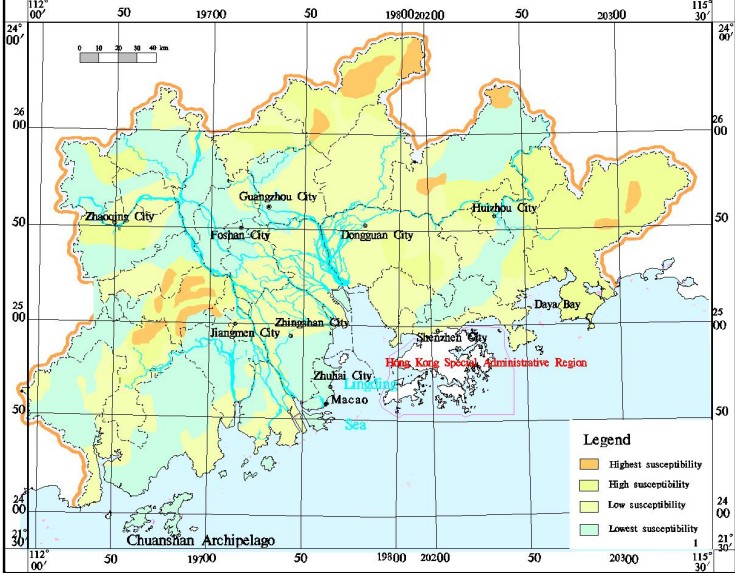


Fig.09 Collapse and landslides susceptibility map of the study area



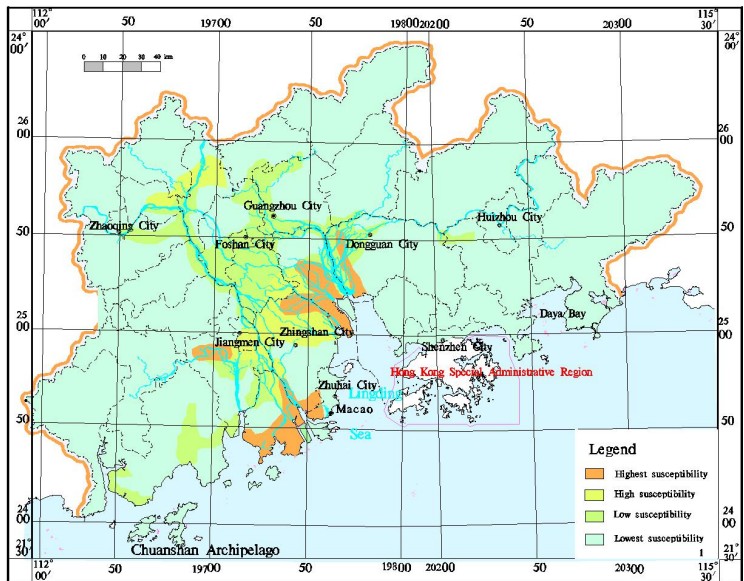


Fig.10 Ground subsidence susceptibility map of the study area

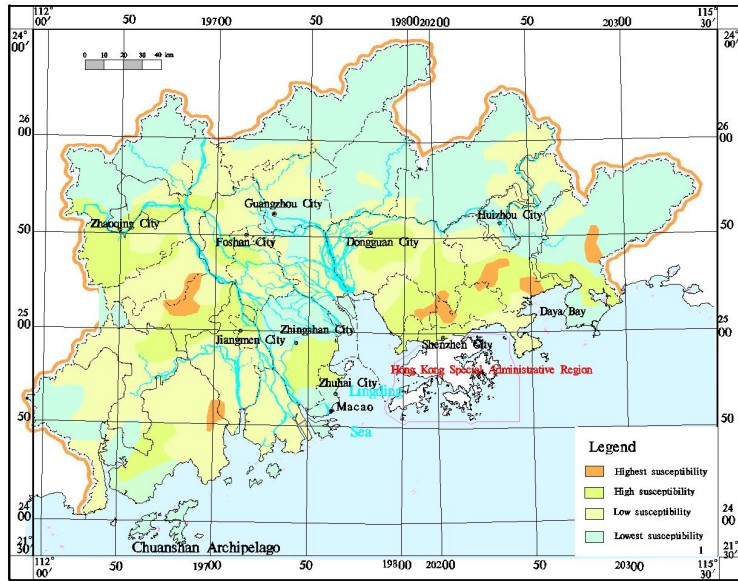


Fig.11 Water and soil erosion susceptibility map of the study area





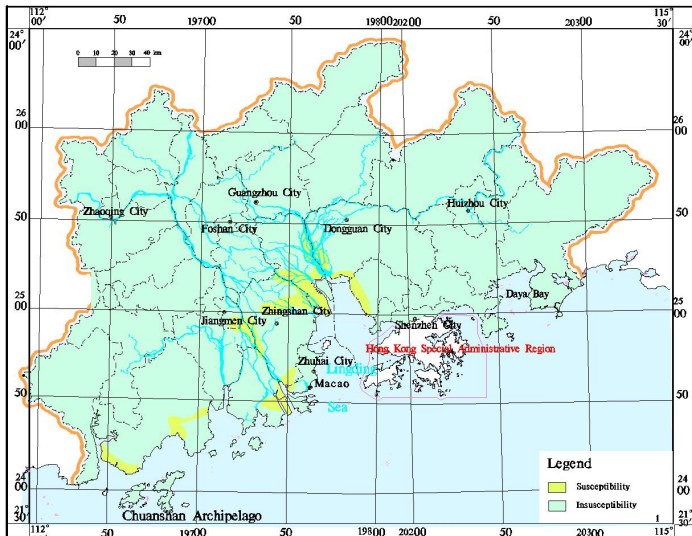

Fig.12 Seawater intrusion susceptibility map of the study area

**4.2 Assessment of multi-hazards susceptibility**
Based on the susceptibility assessment of individual geological hazard, the
multi-hazard susceptibility is evaluated by using the difference method. Moreover, the
multi-hazard susceptibility map for the study area is produced by synthesizing five
geo-hazard maps in the MAPGIS 6.7 platform, and this map was further reclassified
into four classes: high susceptibility, medium susceptibility, low susceptibility and
insusceptible (Fig.13).





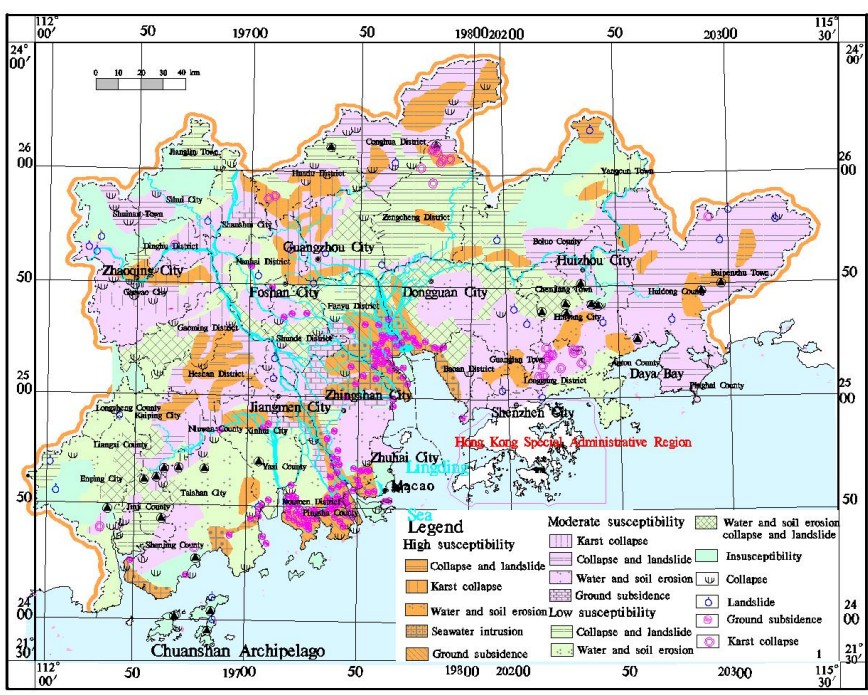


Fig.13 Comprehensive susceptibility map of geological hazards

**4.2.1 High susceptibility zones**
(1) High susceptibility zone of collapses and landslides
This zone is mainly located in the north of Conghua District, Heshan City, the
northern border area of Boluo County and Baipenzhu Town. The zone is mainly
distributed in low mountains and hilly area, which is characterized as steep terrain and
high elevation. The outcropped lithology consists of intrusive rocks and metamorphic
rocks, and metamorphic rocks is characterized as wide distribution, large thickness,
and strong erosion and denudation. Human activities such as slope excavation
contributes to the slope instability under adverse geological conditions. The climate is
complex, with a large annual precipitation, and rainfall is major factor to trigger
geological hazards.
(2)  High susceptibility zone of karst collapse
This zone is mainly located in Huadu District and Nanhai District of Guangzhou City
and Zhaoqing City, few areas of this zone are distributed Boluo County, Huizhou City
and Huidong County. The terrain is relatively flat. This zone is located in hidden karst
areas, so it has the basic conditions for occurrence of karst collapse. So much



infrastructure and large-scale construction projects are built in this zone, and intensity
of human engineering activities is large. Due to much exploiting of groundwater in
the construction of underground engineering, the original balance of rock and soil
mechanics has been artificially changed, causing ground subsidence. The change of
groundwater table is critical factor to trigger geological hazards in this zone.
(3)  High susceptibility zone of water and soil erosion
This zone is mainly distributed in Guanlan Town, Huiyang District, Heshan City and
the eastern area of Taishan City. The engineering geological conditions are complex,
the soil is characterized by loose structure, poor soil viscosity and high erodibility,
especially in Longgang District and Huiyang District, where natural soil erosion is
intense, and the soil are deeply cut by river. Large precipitation, especially heavy rain
and intense rainstorm in the summer, has destroyed the original vegetation and slope
conditions, and has strong erosion and scour on soil.
(4)  High susceptibility zone of seawater intrusion
This zone is mainly distributed in Zhongshan City, Jiangmen City, Nansha District
and Doumen District. This zone is located in delta plain area. A large area of saline
water is formed in this zone, where the salinity of groundwater is high, and seawater
intrusion occurred in part areas. Due to much exploiting of groundwater in the
construction of underground engineering and small annual precipitation, groundwater
cannot be recharged in time, causing lowing of groundwater table which is primary
reason for seawater intrusion to occur in this zone. Moreover, this zone is susceptible
to occur ground subsidence, due to widely distributed mollisol, high water content of
mollisol, and high compressibility of mollisol.
(5)  High susceptibility zone of ground subsidence
This zone is mainly distributed in Fanyu District and Niuwan Town and Pingsha
Town. This zone is located in delta plain area. The outcropped lithology is mainly
sandstone group. The Quaternary sedimentary mollisol with a multi-layer structure
and large thickness is widely distributed and is affected by self-weight, resulting in
self-weight consolidation. So it is prone to ground subsidence.
**4.2.2 Moderate susceptibility zones**
(1) Moderate susceptibility zone of collapses and landslides
This zone is mainly distributed in Conghua District, Nanshui Town, Kaiping City and
the northern area of the study area. This zone is dominated by low mountains and high
hills. The lithology mainly consists of intrusive rocks, volcaniclastic rocks and



metamorphic rocks, with strong erosion. The annual precipitation is large. The rainfall
is the major triggering factor for the occurrence of collapses and landslides.
(2)  Moderate susceptibility zone of water and soil erosion
This zone is mainly distributed in Gaoyao City, Zhaoqing City, Shenjing Town,
Heshan City, Jiangmen City, Zhongshan City area, Dongguan City, Longgang District
and Huiyang District. This zone is mainly dominated by low mountains, hills and
platform. The soil is mostly loamy clay, which is prone to surface loss under lessivage
of clay particles. The outcropped lithology mainly consists of intruded rocks, volcanic
rocks, and layered clastic rocks with carbonate rocks group. The rainfall has strong
erosion and scour on soil. The vegetation coverage is small. The cultivated land is
distributed in the Pearl River Delta coast area, and frequent tillage is more likely to
cause water and soil erosion. The occurrence of water and soil erosion is mainly
triggered by human factors.
(3)  Moderate susceptibility zone of ground subsidence
This zone is mainly distributed in Shunde District, the northwest area of Zhongshan
City, the eastern area of Doumen District and the central area of Sanshui District. This
zone is located in delta plain area. The outcropped lithology is mainly sandstone
group. The mollisol with a multi-layer structure and large thickness is widely
distributed, and the thickness of mollisol range from 5 m to 20 m. Much exploiting of
groundwater causes lowing of groundwater table, resulting in form of depression cone
in exploiting region, which causing compression and consolidation of Quaternary
sand layer. The original balance of rock and soil mechanics has been artificially
changed under human engineering activities, causing ground subsidence.
(4)  Moderate susceptibility zone of karst collapse
This zone is mainly distributed in Dinghu District, the adjacent area between Sihui
City and Shanshui District, Guangdong City, Kaiping City and the northwest marginal
area of Taishan City. This zone is dominated by the delta plain and platform,
characterized by flat terrain and low ground elevation. Engineering geological
conditions is complex, the lithology consists of clastic rock group, red clastic rock
group, and volcanic intrusive rock, with strong erosion. The karst is distributed in
parts area of this zone. Much exploiting of groundwater and mining causes the change
of groundwater table, which is major reason to trigger karst collapse.
**4.2.3 Low susceptibility zones**
(1) Low susceptibility zone of collapses and landslides



This zone is mainly distributed in Conghua District, Boluo County, Jianglin Town,
Fanyu District and Jinji Town. This zone is dominated by low mountains and hills,
and ground elevation is less than 100 m. The outcropped lithology is composed of
intrusive rocks and metamorphic rocks. The engineering geological condition is
simple. The annual precipitation is less than mean annual precipitation for the entire
study area. Thus, it is not prone to collapses and landslides.
(2) Low susceptibility zone of water and soil erosion
This zone is mainly distributed in Liangxi Town, Longsheng Town, Taishan City, Yaxi
Town, Doumen District, Foshan City and Yangcun Town. This zone is dominated by
low mountains and hills. The outcropped lithology consists of sandstone group and
intrusive rocks. This zone is characterized by weak soil erosion, small river system
and large vegetation coverage. Water and soil erosion occurred in few areas of this
zone and is caused by human activities.
(3) Low susceptibility zone of geological hazards
This zone is prone to collapses, landslides, karst collapses and water and soil erosion.
This zone is mainly distributed in Foshan City, the northern area of Dongguan City,
Chenjiang Town, the northern area of Shunde District, and Gaoming District. The
topography consists of low mountains, hills, platform and delta plain. This zone is
characterized by small slope and developed geological structure. But human
engineering activities are weak and precipitation is small. Thus, it is not prone to
trigger geological hazards.
**4.2.3** Insusceptible zone of geological hazards
This zone is mainly distributed in the northwest area of the study area, Enping City,
Shalan Town and Boluo County, it extends for 107 km$^2$, accounting for 8.1% of the
study area. This zone is located in hilly area. The outcropped lithology consists of
metamorphic rocks and intrusive rocks. This zone is characterized by small
population density, large vegetation coverage and weak intensity of human activities,
which has weak destruction on geological environment. Moreover, few geological
hazards are found in this zone and hazards events keep away from residential areas,
which has a weaker threat to the life and property of local residents.
**5 Analysis of the causes of geo-hazards**
**5.1 Composition conditions**
Topography. The geological hazards that are greatly affected by topography are



collapses and landslides within the study area. In the study area, the collapses and
landslides are founded in low mountains and hilly area, karst collapses occur in the
karst development area, water and soil erosion occur in hilly area and platform, and it
mostly occur in the slope with 15° - 30° .
Stratigraphic lithology. The study area is widely distributed with loose
alluvial-diluvial layer, eluvium layer, swell-shrinkage soil and colluvial soil, loose
rocks is characterized by weak lithology and low shear strength. So it is susceptible to
collapses, landslides and other geo-hazards under influence of triggering factors.
Concealed karst is more developed, so it is prone to ground collapse under the action
of human activity. The mollisol is characterized by high water content, high
compressibility, low shear strength and low bearing capacity, so it is prone to ground
subsidence and mollisol foundation subsidence. Weathering residual soil has poor
corrosion resistance, and it is easy to collapse in case of water, so it is prone to water
and soil erosion.
Geologic structure. It is susceptible to cause collapses in some area, characterized by
strong tectonic movement, broken stratum and frequent earthquake.
**5.2 triggering factors**
(1) Precipitation
There are more geological hazards can be found in some areas with large precipitation.
In areas with large annual precipitation, the surface runoff is very strong and the slope
toe are deeply cut by the rivers resulting in formation of temporary surface. The
precipitation can increase pressure of pore water in soil body and reduce the shear
strength of soil body, result that the slope is prone easily destabilized and destructed.
The rainfall for the study area is abundant and has a unevenly temporal distribution.
The raindrop has strong the scouring effect and erosion on ground during rainfall,
resulting in water and soil erosion. Table 3 shows the quarterly distribution
characteristics of collapses during the recent 15 years within the typical area of the
study area. From table 3, it is indicated that collapses primarily occurred in the rainy
season from June to September, and it maintains consistency with the distribution of
monthly precipitation.

<Table 3>





(2) Human activities
Unreasonable human activities are important factors for causing frequent occurrence
of geological hazards such as collapses, landslides, ground subsidence, ground
collapse and so on. In the study area, slope cutting effect under demand of building
houses and road construction have a major impact on the formation of geological
hazard. Human activities such as excavating the slope toe and cutting slope can
change the stress state of the original balance of mountain slope and destroy
vegetation of the slope, so it is easily trigger collapses and landslides. Large-scale
high-rise building construction, exploitation of underground space and other major
projects applied static load on foundation, which can change the stress balance of
engineering foundation and make the soil body of foundation creep, and it cause the
compaction and deformation of soil body. Finally, it will trigger ground collapse and
ground subsidence.
Over-pumping groundwater is primary factor to cause karst collapse and ground
subsidence.

## 6 Conclusions

The aim of this study is to assess multi-hazard susceptibility and identify different
susceptibility area in the Pearl River Delta Economic Region, where various hazards
occurred. This paper presents a first attempt to propose an new method that integrated
the Analytic Hierarchy Process (AHP) and the Difference Method (DM) to assess
multi-hazard susceptility. Based on the geo-hazards investigation and local geological
environmental conditions, this paper systematically analyzes the occurrence
mechanism and formation conditions of geological hazards and summarizes the causal
factors for controlling occurrence of geological hazards. And based on the above
analysis process, individual geo-hazard susceptibility is assessed by applying the
Analytic Hierarchy Process (AHP) and spatial analysis of MapGIS. the multi-hazard
susceptibility is assessed by the Difference Method (DM) based on above individual
geo-hazard susceptibility result, and the assessment results are plotted in a
susceptibility-zoning map of multi-hazard on MapGIS 6.7 platform.
The multi-hazard susceptibility map shows most of areas of the study area are under
the middle and low susceptibility zones, accounting for 75.2% of the toatl study area.
High susceptibility zone covers an area of 6662.24 km$^2$, accounting for 16.5% of the
study area, where geo-hazards are likely to occur due to poor geological environment



and strong human activities. Moderate susceptibility zone covers an area of 16806.91
km$^2$, accounting for 41.6% of the entire study area, remaining area are under low
susceptibility zone and insusceptible zone, accounting for 41.9% of the entire study
area. From multi-hazard susceptibility map, geological hazards events are distributed
in corresponding susceptibility zone, which verifies the accuracy of new method and
indicated that this method suits the study area. This study can provide theoretical
guide to urban planning and geo-hazards management for achieving the optimal
allocation of geological resources and environment, and it can be combined with
present land-use map to provide scientific basis to adjust land use planning, coming
true the rational use of land resources.





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





**Table 1 Value of RI**

| Order n | 1 | 2 | 3 | 4 | 5 | 6 | 7 | 8 | 9 |
|---|---|---|---|---|---|---|---|---|---|
| RI | 0.00 | 0.00 | 0.58 | 0.90 | 1.12 | 1.24 | 1.32 | 1.41 | 1.45 |


**Table 2**
**Assessment factor system of geological hazards susceptibility**

| Criterion layer | Evaluation index | The score and rank of assessment indexes | | | | Weight |
|---|---|---|---|---|---|---|
| | | 1 | 2 | 3 | 4 | |
| Karst collapse | Degree of karst development | Strong | Moderate | Poor | — | 0.2100 |
| | Overburden thickness (m) | <10 | 10-30 | >30 | — | 0.3211 |
| | Lithology | Limestone,dolomite | Glutenite,mud limestone,tuff, sandstone | Clay rock, mudstone, shale, silty sandstone, silty slate | — | 0.2100 |
| | Aquifer water yield property | Weak | Moderate | rich | — | 0.1001 |
| | The distance to the fracture | 0-2000 | 2000-4000 | >4000 | — | 0.1587 |
| Collapse and landslide | Topography | Delta plain,marine deposition terrace | Alluvial plains, alluvial and diluvial plains, alluvial and marine deposition plains | Hilly area | Low mountainous area | 0.3300 |
| | Lithology | Pluton, shale | Medium - thick layered carbonate rocks, | layered metamorphc rock | layered clastic rocks | 0.3300 |
| | The distance to faults | <1000 | 1000-2000 | 2000-3000 | >3000 | 0.1996 |
| | Precipitation | <1600 | 1600-1800 | 1800-2000 | >2000 | 0.1404 |
| Ground subsidence | The thickness of deposition | <10 | 10～20 | >20 | — | 0.4249 |
| | Aquifer water yield property | Weak | moderate | rich | — | 0.2701 |
| | The deposition age of millisol | Holoce nemarine deposit - sea alluvial Guizhou group, the upper Pleistocene middle | Holocene alluvia Dawan Town group - the middle Holocene lake marsh Mugao group | Holocene alluvial group, red bed residual soil | — | 0.1613 |
| | The distance to the fracture | <2000 | 2000-4000 | >4000 | — | 0.1438 |
| Water and soil erosion | Topography | Delta plain,marine deposition terrace | Alluvial Plains, alluvial and diluvial plains, alluvial and marine deposition plains | Hilly area | Low mountainous area | 0.2140 |
| | Vegetation type | Arbor,shrub | Economic forest, shelterbelt, | crops | Unused land | 0.2499 |
| | Soil type | Paddy soil | Red loam | Fluvo-aquic soil | Latosolic red soil | 0.3079 |
| | Precipitation | <1800 | 1800-2000 | 2000-2200 | >2200 | 0.1191 |
| | the density of of river network | Scatted | More scattered | Even | Concentrated | 0.1092 |

| seawater intrusion | Topography | Delta plain, marine deposition terrace | Alluvial plains, alluvial and diluvial plains | hilly area | Low mountainous area | 0.1438 |
|---|---|---|---|---|---|---|
| | The type of Quaternary sedimentary rock | Bedrock | Holocene lacustrine sediment colluvium | Holocene marine clay | Holocene sea alluvial clay | 0.1613 |
| | Groundwater table | <0 | 0-10 | 10-60 | >60 | 0.2701 |
| | Precipitation | <1800 | 1800-2000 | 2000-2200 | >2200 | 0.4249 |


**Table 3**
**The quarterly distribution characteristics of collapses in the study area between 1990 and**
**2006**

| Time | Jan - March | | Apr - Jun | | July - Sep | | Oct - Dec | |
|---|---|---|---|---|---|---|---|---|
| City | Number | Percentage | Number | Percentage | Number | Percentage | Number | Percentage |
| Zhaoqing City | 13 | 6.88 | 51 | 26.98 | 97 | 51.32 | 28 | 14.82 |
| Huizhou City | 3 | 3.06 | 22 | 22.45 | 61 | 62.25 | 12 | 12.25 |
| Guangzhou City | 11 | 3.77 | 102 | 34.93 | 162 | 55.48 | 17 | 5.82 |
| Shenzhen City | 19 | 4.97 | 60 | 15.71 | 278 | 72.78 | 25 | 6.55 |
