# Peer review of "The susceptibility assessment of multi-hazard in the Pearl"

_Natural Hazards and Earth System Sciences, 2018_

## Referee Comment (RC1) · Anonymous Referee #1 · 20 Jun 2018

In this study, the authors presented an approach based on Analytic Hierarchy Process (AHP) and the Difference Method (DM) to assess multi-hazard susceptibility to identify susceptibility areas in the Pearl River Delta region. This is a cutting-edge idea to find geological hazard susceptibility areas, which is quite import and valuable for decision-making and geological hazard management. However, the paper has several crucial faults so that I don't think the current version is good enough to publish in this journal. I leave the decision to the editor to decide the fate of the manuscript. The first major problem is, the authors did not provide / collect enough data to support their study which make the results based on the inadequate data unbelievable. For example, the authors only presented datasets of precipitation, topography, geology,

etc. Such datasets are enough to assess geological hazard susceptibility? Obviously, no. For instance, soil erosion and landslides have a strong relationship with vegetation cover; without vegetation cover map, how the authors could provide a correct susceptibility map for landslides or soil erosion? Seawater intrusion also has a strong relationship with freshwater discharge provided by the Pearl River, without hydrological data (e.g., annual mean runoff and long-term runoff fluctuation; besides, insufficient seasonal runoff caused by human activities (irrigation, water impoundment for energy generation) also can led to seawater intrusion), is it possible to identify regions with high susceptibility to seawater intrusion? The second problem, which is the key issue, is that the results are questionable. As we all know, for each geological hazard listed in this study, there is susceptibility map already. We can just compare the results in the study with the maps released by government or previous studies. For example:Zhao et al. (2014) released susceptibility map for landslides for the whole Guangdong Province (obviously include the entire delta area of the Pearl River): (Figure from Zhao et al. (2014), please see the attached pdf file) The authors' result failed to identify the circled area as the high susceptibility area for landslides.

According to the investigation by Geological Survey Bureau of China (http://www.cigem.cgs.gov.cn/cgkx_4859/201703/t20170316_424756.html), the high susceptibility areas for ground subsidence in the Pearl River Delta area are Foshan, Guangzhou, Jiangmen, Zhongshan, Zhuhai and Shenzhen. However, the result map provided by the authors didn't identify Shenzhen as the high susceptibility area. In fact, ground subsidence events in Shenzhen have been reported by many studies.

For soil erosion susceptibility map, we can also get the soil erosion map for the Guangdong Province via the link (http://www.dsac.cn/DataProduct/Detail/20080604) (please see the attached pdf file)

The soil erosion map provided by the study also failed to identify the circled area as the high susceptibility area for landslides (please see the attached pdf file).

All in all, I think the results provided by the authors are quite unbelievable.

The third problem is that the manuscript has too many English grammatical mistakes, the authors MUST carefully check each sentences before next submission. For example (in first six pages I can find near 20 problems): Line 13: The main scope of this paper [is assess]multi-hazard susceptibility to identify area by using an integrated susceptibility. . . Line 29: geo-hzards management Line 57: loss of human life (lives?), reduce economic consequences (or loss?) Line 58: it is very meaning (meaningful?) to Line 61: Since geological hazards are (a?) complex phenomena Line 65: relative information of different hazards is (an?) important tool Line 69: a complex process and confronted with a challenges Lines 73-74: describe the real relationships of different influencing factors (why you describe the relationships between different influencing factors?, you should investigate the relationship between result with the influencing factors). Line 77: One widely used method of (for?) susceptibility assessment Line 79: hazards susceptibility in this (a ?) unit is considered high Line 91: hazard susceptibility is assessed [with via??] of the Analytic Hierarchy Process Line 93: The difference (different?) method is used Line 94: the five aforementioned geohazards (geohazard?) susceptibility assessment Line 109: Fig.1 The map (what map?) of the study area in The Pearl River Delta Economic Zone Line 112: The rainfall is characterized by large (high?) precipitation Line 117: Fig.2 The (spatial distribution of?) precipitation map of the study area Line 121: The terrain is smooth (flat?), Line 122: Based on the different genetic (what do you mean genetic? Geological?) type Reference: ZHAO HongtingïijŇLIU XilinïijŇYU ChengjunïijŇSHANG ZhihaiïijŐRisk Assessment and Temporal-Spatial Changes of CollapseïijŇLandslide and Debris Flow in Guangdong[J]ïijŐTropical GeographyïijŇ2014ïijŇ34ïijĹ6ïijĽïijŽ804-813ïijŐ

Please also note the supplement to this comment:
https://www.nat-hazards-earth-syst-sci-discuss.net/nhess-2018-104/nhess-2018-104-RC1-supplement.pdf

---

## Referee Comment (RC2) · Anonymous Referee #2 · 22 Jun 2018

The manuscript describes an attempt to perform a multi-hazard susceptibility map of Pearl Delta Economic Zone in China. The issue of multi-hazard assessment is very interesting and it has still many open questions. The term "multi-hazard" is frequently used in the literature as an adjective to indicate multiple sources of hazard that are analysed in parallel and finally integrated into a multi-risk analysis. According to Corominas et al. (2013) multi-hazard assessment should refer to the joint probability of independent events occurring in the same area in a given time span. Multi-hazard assessment becomes relevant when hazard sources can interact, giving rise to a domino effect that occurs when a hazard event triggers a secondary event. In this paper six different hazards are considered. For each of them a susceptibility map is performed and then

all the six maps are simply overlaid in GIS enviroment, providing a final multi-hazard map, thus not considering any potential interactions among the hazards and the possible domino effects. For example landslides ad soil erosion are stricltly connected and this aspect should be considered in the analysis. Furthermore the susceptibility assessment of each specific hazard is carried out with a simple method, completely neglecting the extensive literature on the geohazards susceptibility assessment, with special reference to the methodology and the selection and preparation of predisposing factors. In particular the selection of predisposing factors is arguable and an incomplete set of factors is considered. Some important factors, infact (such for example vegetation) are not considered. Another relevant limitation of the analysis is that no information are provided about the model input data (resolution, date, source). In particular the resolution of the input data affects the resolutions of the final susceptibility maps that are not provided in the work. Eventually, what about the geological hazards database (landslide inventory map, collapse map and so on..) used to assess susceptibility? Some detailed information should be provided.

Additionally to these general considerations, I have several specific comments, listed below:

- The introductuion is too long and not well focused. Some parts are useless (lines 37-50) and some other parts are not clear (lines 85-88). It is quite curious that in the Introduction you don't mention which type of hazards you consider in your analysis. In line 94 you state "...aforementioned geohazards" but I cannot find where you have mentioned them!; - Some sections inside the study area are very short (2.2, 2.3 and 2.4). I think they can be merged; - What is the definition of collapse in your work and which is the difference betwen collapse and karst collapse? Furthemore why landslide and collapse have the same casual factors? In my point of view they are quite different phenomena. This point shoud be clarified better. - No quantitative validation of susceptibility maps is performed. This is not correct since every model has to be validated in order to evaluate its performance. There is only one general sentence in the Conclusion (lines 566-568), which is absolutly not enough. - Figures are too small, words inside are not redable. - In general English is not good. Even tough I am not an English mother tongue, I have identified several errors and mistakes and the authors have to carefully check the language.

Based on the above comments I think that the manuscript cannot be accepted for publication in the journal. My main concerns, as stated above, are related to the methodological approach that is too superficial and do not take into account relevant aspects related to susceptibility assessment.

---

## Author Comment (AC1) · 7 Jul 2018

Comments of referee 1: the authors did not provide/collect enough data to support their study which make the results based on the inadequate data unbelievable. For example, the authors only presented datasets of precipitation, topography, geology etc. Such datasets are enough to assess geological hazard susceptibility? Obviously, no. For instance, soil erosion and landslides have a strong relationship with vegetation cover; without vegetation cover map, how the authors could provide a correct susceptibility map for landslides or soil erosion? Author's response: It is our negligence that some data used to assess geo-hazards susceptibility are not presented in

this paper. According to your kind suggestion, we have now added data in section 2 to support this study. Author's changes in manuscript: We have provided the spatial distribution map of vegetation type (Fig.08) in line 206. Vegetation coverage is shown in Fig.07. In addition, we have add fracture data in Fig.04. Due to the limitations of coverage, other data are provided in supplement data. Fracture information has been added in section 2.2 in as following: "The Pearl River Delta belongs to the South China fold belts, the Northern and Central Guangdong depression belt, and the main depression is the Sanshui Depression Basin. Some large fractures develop in the study area (Fig.04). These large faults are characterized by multiple phases of activity, especially since the late Tertiary, and they have affect on formation and evolution of The Pearl River Delta, structure development and crustal stability." Comments of referee 1: Seawater intrusion also has a strong relationship with freshwater discharge provided by the Pearl River, without hydrological data (e.g., annual mean runoff and long-term runoff fluctuation; besides, insufficient seasonal runoff caused by human activities (irrigation, water impoundment for energy generation) also can led to seawater intrusion), is it possible to identify regions with high susceptibility to seawater intrusion? Author's response: We fully agree with the referee on the effect of above hydrological data on seawater intrusion. However, we select influencing factors to assess seawater intrusion susceptibility based on occurrence mechanism and formation conditions of seawater intrusion, and we consider basic formation conditions of seawater intrusion as assessment indexes. So, hydrological data (e.g., annual mean runoff and long-term runoff fluctuation) and insufficient seasonal runoff are not considered in this paper, the type of Quaternary sedimentary rock only is considered in seawater susceptibility assessment. The type of Quaternary sedimentary rock affects the hydraulicÂărelation between the seawater and the land aquifer, so it affects the occurrence of seawater intrusion. Comments of referee1:The second problem, which is the key issue, is that the results are questionable. As we all know, for each geological hazard listed in this study, there is susceptibility map already. We can just compare the results in the study with the maps released by government or previous studies. For example:

Zhao et al. (2014) released susceptibility map for landslides for the whole Guangdong Province (obviously include the entire delta area of the Pearl River): (Figure from Zhao et al. (2014), please see the attached pdf file) The authors' result failed to identify the circled area as the high susceptibility area for landslides. Author's response: It is our negligence that the range of the study area is not clarified in the paper. The circled high susceptibility area from Zhao et al. (2014) is located in Qingyuan City, which does not belong to the Pearl River Delta Economic Zone. Zhao et al. (2014) took cities at county level as evaluation units, which affected the accuracy of susceptibility assessment. Moreover, Zhao et al. (2014) carried out collapse, landslide and debris flow hazards susceptibility assessment based on the existing methodology for debris flow hazard assessment, resulting that the result provided Zhao et al. (2014) has a slight difference with our result of collapse and landslide susceptibility. Comments of referee 1: According to the investigation by Geological Survey Bureau of China (http://www.cigem.cgs.gov.cn/cgkx_4859/201703/20170316_424756.html),the high susceptibility areas for ground subsidence in the Pearl River Delta area are Foshan, Guangzhou, Jiangmen, Zhongshan, Zhuhai and Shenzhen. However, the result map provided by the authors didn't identify Shenzhen as the high susceptibility area. In fact, ground subsidence events in Shenzhen have been reported by many studies. Author's response: We carried out ground subsidence susceptibility assessment based on the consideration of occurrence mechanism and formation conditions of ground subsidence. The thinner mollisol is distributed in Shenzhen City, and the intensity of groundwater development and utilization is small, the probability of formation of ground subsidence is small. Lu et al. (2006) released geological disaster susceptibility map through the research on the distribution of geological environment conditions and the effecting factors of the geological disasters. The result provided by Lu et al. (2006) shows that ground subsidence is not prone to occur in Shenzhen City, which is consistent with our study result. Shenzhen City are under increasing urbanization and development intensity, resulting that load on the ground trends to increase, which has a effect occurrence of ground subsidence. It is limitation of our study that urban development is not considered in ground subsidence susceptibility assessment. Comments of referee 1: For soil erosion susceptibility map, we can also get the soil erosion map for the Guangdong Province via the link (http://www.dsac.cn/DataProduct/Detail/20080604) (please see the attached pdf file) The soil erosion map provided by the study also failed to identify the circled area as the high susceptibility area for landslides (please see the attached pdf file). All in all, I think the results provided by the authors are quite unbelievable. Author's response: It is our negligence that the range of the study area is not clarified in the paper. The Pearl River Delta Economic Zone includes Guangzhou City, Shenzhen City, Foshan City, Zhongshan City, Huizhou City, Dongguan City, Zhuhai City, Jiangmen City, Zhaoqing City. The circled high susceptibility area in the soil erosion map (shown the attached pdf file) is located in Shaoguan City, So it does not belong to the Pearl River Delta Economic Zone. Comments of referee 1: the third problem is that the manuscript has too many English grammatical mistakes, the authors MUST carefully check each sentences before next submission. For example (in first six pages I can find near 20 problems): Author's response: According to your kind suggestion, we have corrected English grammatical mistakes in first six Page, the revised portions were marked in red as following. And, we carefully have checked and corrected all language of this paper. Author's changes in manuscript: Grammatical mistakes in lineÂă13 has been corrected as following:Âă"TheÂămainÂăscopeÂăofÂăthisÂăpaper is to assess multi‐hazardÂă-susceptibilityÂăandÂăidentifyÂăareaÂăby usingÂăanÂăintegratedÂăsusceptibility…". LineÂă29:Âă"geo‐hzardsÂămanagement" has been corrected as "geo-hazards management". LineÂă57:ÂăThis sentence has been corrected as "in order to minimize the economic loss and reduce threaten on people's lives and property" in now line 52-53. LineÂă58:Âă"Meaning" has been corrected as "meaningful" in now line 54. Line 61: This sentence has been corrected as "Since geological hazards are a phenomena" in now line 56. Line 65: This sentence has been corrected as "relation between different hazards is an important tool " in now line 59-61. Grammatical mistakes in line 69 has been corrected as "a complex process and confronts

with challenges" in now line 64. LinesÂă73‐74:ÂăThis sentence has been corrected as following: "which are based on statistical analysis of the scale and density of occurred geological hazards, but it is difficult to describe the affect of different influencing factors on occurred geological hazards" in now line 67-69. Grammatical mistakes in line 77 has been corrected as "OneÂăwidelyÂăusedÂămethodÂăfor susceptibilityÂăassessment" in now line 72. Grammatical mistakes in line 77 has been corrected as "hazardsÂăsusceptibilityÂăinÂăthisÂăaÂăÂăunitÂăisÂăconsideredÂăhigh" Âăin now line 82. LineÂă91:ÂăLanguage of this sentence is right. LineÂă93:Âă"TheÂădifferenceÂămethod" has been corrected as "the Difference Method" in now line 89. The basic principle of the Difference Method is CannikinÂăLaw. Grammatical mistakes in line 94 has been corrected "the five fiveÂăaforementionedÂăgeohazard susceptibility assessments" in now line 89-90. Line 109: the caption of Fig.1Âăhas been corrected as "Fig.1ÂăTheÂăspatialÂădistributionÂămap of geo-hazards in the study area" in now line 104. Grammatical mistakes in line 112 has been corrected as " The rainfall is characterized by high precipitation" in now line 107. The caption of fig.2 has been corrected as "Fig.2ÂăTheÂăspatialÂădistributionÂămap ofÂăprecipitationÂăofÂătheÂăstudyÂăarea" in now line 113. LineÂă121:ÂăÂă"smooth"Âăhas been corrected asÂă"flat" in now line 117. LineÂă122: Genetic type can be understand as origin type in geology.
* * *
**Fig. 1.**

Fig. 2.

---

## Author Comment (AC3) · 7 Jul 2018

Comments of referee 2: The manuscript describes an attempt to perform a multi-hazard susceptibility map of Pearl Delta Economic Zone in China. The issue of multi-hazard assessment is very interesting and it has still many open questions. The term "multi-hazard" is frequently used in the literature as an adjective to indicate multiple sources of hazard that are analysed in parallel and finally integrated into a multi-risk analysis. According to Corominas et al. (2013) multi-hazard assessment should refer to the joint probability of independent events occurring in the same area in a given time span. Multi-hazard assessment becomes relevant when hazard sources can interact, giving rise to a domino effect that occurs when a hazard event triggers a secondary event. In this paper six different hazards are considered. For each of them a susceptibility map is performed and then all the six maps are simply overlaid in GIS environment, providing a final multi-hazard map, thus not considering any potential interactions among the hazards and the possible domino effects. Comments of referee 2: For example landslides ad soil erosion are strictly connected and this aspect should be considered in the analysis. Author's response: We selected influencing factors affecting occurrence mechanism and formation conditions of individual geo-hazaras as susceptibility indexes. Multi-hazard susceptibility assessment is carried out by using the Difference Method to provide decision-makers with visual information on the spacial distribution of various geological disasters susceptibility for geo-hazards management and land use planning, which reduce confusion of decision-makers on high number of individual geo-hazard maps. Collapse and landslide have theÂăinherentÂărelationship in occurrence mechanism and formation condition, so we carry out collapse and landslide susceptibility assessment. And, there are weak correlation between other geo-hazards in occurrence mechanism and formation condition. Thus, the relation between geo-hazards is considered in geo-hazards susceptibility assessment. Comments of referee 2: Furthermore the susceptibility assessment of each specific hazard is carried out with a simple method, completely neglecting the extensive literature on the geohazards susceptibility assessment, with special reference to the methodology and the selection and preparation of predisposing factors. In particular the selection of predisposing factors is arguable and an incomplete set of factors is considered. Some important factors, infact (such for example vegetation) are not considered. Author's response: The Analytic Hierarchy Process verified by lots of researches is widely used to evaluate geo-hazards susceptibility. Geo-hazards susceptibility assessment is carried out by using AHP and GIS, which reduce obvious subjectivity and provide visual information. And, we have performed validation of susceptibility results through sensitivity analysis. The sensitivity analysis result indicates the accuracy and rationality of the assessment method and shows it suits the particularity of the study area. We selected

influencing factors as individual geo-hazard susceptibility indexes based on consideration for occurrence mechanism and formation conditions of individual geo-hazard, human triggering factors were not considered. Comments of referee 2: Another relevant limitation of the analysis is that no information are provided about the model input data (resolution, date, source). In particular the resolution of the input data affects the resolutions of the final susceptibility maps that are not provided in the work. Eventually, what about the geological hazards database (landslide inventory map, collapse map and so on..) used to assess susceptibility? Some detailed information should be provided. Author's response: We took into account this comment. Author's changes in manuscript: We have added a new chapter (section 3.1) to provided some information about data. High number of data have been used in geo-hazard susceptibility assessment, along of them, natural geographical conditions, geological condition, the spacial distributed of major geological hazards, vegetation cover and land use type have been provided in section 2, soil type, the distribution and thickness of mollisol, and groundwater information is provided in supplement data. Comments of referee 2: The introduction is too long and not well focused. Some parts are useless (lines 37-50) and some other parts are not clear (lines 85-88). It is quite curious that in the Introduction you don't mention which type of hazards you consider in your analysis. In line 94 you state "...aforementioned geohazards" but I cannot find where you have mentioned them! Author's response: We took into account this comment. Author's changes in manuscript: For content in lines 37-50, we only have removed lines 43-50, because the research background was introduced in line 37-43. To clarify the statements, we have rewritten the paragraph line 85-88 (new line 79-83) in the following way: "Thus, multi-hazard susceptibility assessment is completed by synthesizing all individual geological hazards susceptibility result with the Difference Method. The major principle of the Difference Method is that the multi-hazard susceptibility in this a unit is considered high, as long as there is a kind of geological hazard under high susceptibility in specific evaluation unit. And, we have clarified type of hazards occurred in the study area in line 48-52: According to geological survey result, geo-hazards occurred

in the Pearl River Delta Economic Zone mainly include collapse, landslide, debris flow, ground subsidence, karst collapse, water and soil erosion and seawater intrusion, the scale of debris flow is small, so it is not considered as object of study." Comments of referee 2: Some sections inside the study area are very short (2.2, 2.3 and 2.4). I think they can be merged. Author's response: According to your kind suggestion, section 2.2, 2.3 and 2.4 have been merged in section 2.2. Original section 2.2, 2.3 and 2.4 are now section 2.2.1, .2.2.2 and 2.2.3, respectively. Comments of referee 2: What is the definition of collapse in your work and which is the difference between collapse and karst collapse? Author's response: Collapse refers that rock-soil block is divorced from the slope body under the influence of external factors. Collapse belongs to the problem of slope rock mass instability. When natural caves is affected by sudden changes of stability conditions due to karst dissolution, karst collapse trends to occur. The formation conditions of karst collapse mainly includes karst with certain development, overlying rock and soil, and the karst groundwater system. Comments of referee 2: Furthemore why landslide and collapse have the same casual factors? In my point of view they are quite different phenomena. This point shoud be clarified better. Author's response: Collapse and landslide belong to the problem of slope rock mass instability, they often associated with each other in the cause of formation. There are also internal relations and transformation relations between them, which make them have strong consistency in temporal and spatial distribution. Landslides are closely related with collapses, and they usually occur accompanied. Collapse and landslide occur under the same geological tectonic setting and the same stratigraphic lithology conditions, with the same triggering factors. So they have the same casual factors. Author's changes in manuscript: We also clarified reason why landslide and collapse have the same causal factors in line 234-242. Comments of referee 2: No quantitative validation of susceptibility maps is performed. This is not correct since every model has to be validated in order to evaluate its performance. Author's response: We have carried out quantitative validation of susceptibility maps by sensitivity analysis (analysis of effective weight). The sensitivity analysis result validates the accuracy and rationality of

the assessment method and shows it suits the particularity of the study area. Author's changes in manuscript: Validation ofÂătheÂăresults is added in section 6. The text as been added as followed: "Sensitivity analysis is used to assess effects of the input criteria on the model output performance and also to validate the effect of changing variable conditions or parameter values on the system (Gomez and Jones 2010). Sensitivity analysis is to determine the effective weight of each parameter and compared it with the theoretical weight for verifying the information on the effect of scaled values and weights assigned to each parameter. The effective weight, called coefficient of variation, is computed using the following Eq.Âă(4) (Napolitano and Fabbri 1996).

where W is the effective weight of the parameter P, Pr and Pw are the weight and the scaled value of the parameter, respectively, and V is the susceptibility index of geohazard. The result of individual geo-hazard susceptibility assessment (e.g. collapse and landslide, Karst collapse, ground subsidence, water and soil erosion and seawater intrusion) is verified by sensitivity analysis. The sensitivity analysis result is shown in Table 4. Table 4 represents the effective weights with the theoretical weight for individual geo-hazard susceptibility criterion. The effective weights of each parameters are slightly different from the theoretical weight assigned to individual geo-hazard susceptibility, which validates the accuracy and rationality of the assessment method." Comments of referee 2: There is only one general sentence in the Con-clusion (lines 566-568), which is absolutely not enough. Author's response: We do not understand this comment well, so we hope you can give more comments. Comments of referee 2: Figures are too small, words inside are not redable. Author's response: The size of figures and the formatting of words inside, such as font, font size, and spacing have been adjusted for redable. The resolution of figures has been raised to be visible clearly. Comments of referee 2: In general English is not good. Even tough I am not an English mother tongue, I have identified several errors and mistakes and the authors have to carefully check the language. Author's response: we carefully have checked all language of this paper and corrected errors and mistakes.

Please also note the supplement to this comment:
https://www.nat-hazards-earth-syst-sci-discuss.net/nhess-2018-104/nhess-2018-104-AC3-supplement.zip

---

## Author Comment (AC4) · 7 Jul 2018

Comments of referee 2: The manuscript describes an attempt to perform a multi-hazard susceptibility map of Pearl Delta Economic Zone in China. The issue of multi-hazard assessment is very interesting and it has still many open questions. The term "multi-hazard" is frequently used in the literature as an adjective to indicate multiple sources of hazard that are analysed in parallel and finally integrated into a multi-risk analysis. According to Corominas et al. (2013) multi-hazard assessment should refer to the joint probability of independent events occurring in the same area in a given time span. Multi-hazard assessment becomes relevant when hazard sources can interact, giving rise to a domino effect that occurs when a hazard event triggers a secondary event. In this paper six different hazards are considered. For each of them a susceptibility map is performed and then all the six maps are simply overlaid in GIS environment, providing a final multi-hazard map, thus not considering any potential interactions among the hazards and the possible domino effects. Comments of referee 2: For example landslides ad soil erosion are strictly connected and this aspect should be considered in the analysis. Author's response: We selected influencing factors affecting occurrence mechanism and formation conditions of individual geo-hazaras as susceptibility indexes. Multi-hazard susceptibility assessment is carried out by using the Difference Method to provide decision-makers with visual information on the spacial distribution of various geological disasters susceptibility for geo-hazards management and land use planning, which reduce confusion of decision-makers on high number of individual geo-hazard maps. Collapse and landslide have theÂãinherentÂãrelationship in occurrence mechanism and formation condition, so we carry out collapse and landslide susceptibility assessment. And, there are weak correlation between other geo-hazards in occurrence mechanism and formation condition. Thus, the relation between geo-hazards is considered in geo-hazards susceptibility assessment. Comments of referee 2: Furthermore the susceptibility assessment of each specific hazard is carried out with a simple method, completely neglecting the extensive literature on the geohazards susceptibility assessment, with special reference to the methodology and the selection and preparation of predisposing factors. In particular the selection of predisposing factors is arguable and an incomplete set of factors is considered. Some important factors, infact (such for example vegetation) are not considered. Author's response: The Analytic Hierarchy Process verified by lots of researches is widely used to evaluate geo-hazards susceptibility. Geo-hazards susceptibility assessment is carried out by using AHP and GIS, which reduce obvious subjectivity and provide visual information. And, we have performed validation of susceptibility results through sensitivity analysis. The sensitivity analysis result indicates the accuracy and rationality of the assessment method and shows it suits the particularity of the study area. We selected influencing factors as individual geo-hazard susceptibility indexes based on consideration for occurrence mechanism and formation conditions of individual geo-hazard, human triggering factors were not considered. Comments of referee 2: Another relavant limitation of the analysis is that no information are provided about the model input data (resolution, date, source). In particular the resolution of the input data affects the resolutions of the final susceptibility maps that are not provided in the work. Eventually, what about the geological hazards database (landslide inventory map, collapse map and so on..) used to assess susceptibility? Some detailed information should be provided. Author's response: We took into account this comment. Author's changes in manuscript: We have added a new chapter (section 3.1) to provided some information about data. High number of data have been used in geo-hazard susceptibility assessment, along of them, natural geographical conditions, geological condition, the spacial distributed of major geological hazards, vegetation cover and land use type have been provided in section 2, soil type, the distribution and thickness of mollisol, and groundwater information is provided in supplement data. Comments of referee 2: The introduction is too long and not well focused. Some parts are useless (lines 37-50) and some other parts are not clear (lines 85-88). It is quite curious that in the Introduction you don't mention which type of hazards you consider in your analysis. In line 94 you state "...aforementioned geohazards" but I cannot find where you have mentioned them! Author's response: We took into account this comment. Author's changes in manuscript: For content in lines 37-50, we only have removed lines 43-50, because the research background was introduced in line 37-43. To clarify the statements, we have rewritten the paragraph line 85-88 (new line 79-83) in the following way: "Thus, multi-hazard susceptibility assessment is completed by synthesizing all individual geological hazards susceptibility result with the Difference Method. The major principle of the Difference Method is that the multi-hazard susceptibility in this a unit is considered high, as long as there is a kind of geological hazard under high susceptibility in specific evaluation unit. And, we have clarified type of hazards occurred in the study area in line 48-52: According to geological survey result, geo-hazards occurred in the Pearl River Delta Economic Zone mainly include collapse, landslide, debris flow, ground subsidence, karst collapse, water and soil erosion and seawater intrusion, the scale of debris flow is small, so it is not considered as object of study." Comments of referee 2: Some sections inside the study area are very short (2.2, 2.3 and 2.4). I think they can be merged. Author's response: According to your kind suggestion, section 2.2, 2.3 and 2.4 have been merged in section 2.2. Original section 2.2, 2.3 and 2.4 are now section 2.2.1, .2.2.2 and 2.2.3, respectively. Comments of referee 2: What is the definition of collapse in your work and which is the difference between collapse and karst collapse? Author's response: Collapse refers that rock-soil block is divorced from the slope body under the influence of external factors. Collapse belongs to the problem of slope rock mass instability. When natural caves is affected by sudden changes of stability conditions due to karst dissolution, karst collapse trends to occur. The formation conditions of karst collapse mainly includes karst with certain development, overlying rock and soil, and the karst groundwater system. Comments of referee 2: Furthemore why landslide and collapse have the same casual factors? In my point of view they are quite different phenomena. This point shoud be clarified better. Author's response: Collapse and landslide belong to the problem of slope rock mass instability, they often associated with each other in the cause of formation. There are also internal relations and transformation relations between them, which make them have strong consistency in temporal and spatial distribution. Landslides are closely related with collapses, and they usually occur accompanied. Collapse and landslide occur under the same geological tectonic setting and the same stratigraphic lithology conditions, with the same triggering factors. So they have the same casual factors. Author's changes in manuscript: We also clarified reason why landslide and collapse have the same causal factors in line 234-242. Comments of referee 2: No quantitative validation of susceptibility maps is performed. This is not correct since every model has to be validated in order to evaluate its performance. Author's response: We have carried out quantitative validation of susceptibility maps by sensitivity analysis (analysis of effective weight). The sensitivity analysis result validates the accuracy and rationality of the assessment method and shows it suits the particularity of the study area. Author's changes in manuscript: Validation ofÂătheÂăresults is added in section 6. The text as been added as followed: "Sensitivity analysis is used to assess effects of the input criteria on the model output performance and also to validate the effect of changing variable conditions or parameter values on the system (Gomez and Jones 2010). Sensitivity analysis is to determine the effective weight of each parameter and compared it with the theoretical weight for verifying the information on the effect of scaled values and weights assigned to each parameter. The effective weight, called coefficient of variation, is computed using the following Eq.Âă(4) (Napolitano and Fabbri 1996).

where W is the effective weight of the parameter P, Pr and Pw are the weight and the scaled value of the parameter, respectively, and V is the susceptibility index of geohazard. The result of individual geo-hazard susceptibility assessment (e.g. collapse and landslide, Karst collapse, ground subsidence, water and soil erosion and seawater intrusion) is verified by sensitivity analysis. The sensitivity analysis result is shown in Table 4. Table 4 represents the effective weights with the theoretical weight for individual geo-hazard susceptibility criterion. The effective weights of each parameters are slightly different from the theoretical weight assigned to individual geo-hazard susceptibility, which validates the accuracy and rationality of the assessment method." Comments of referee 2: There is only one general sentence in the Con-clusion (lines 566-568), which is absolutely not enough. Author's response: We do not understand this comment well, so we hope you can give more comments. Comments of referee 2: Figures are too small, words inside are not redable. Author's response: The size of figures and the formatting of words inside, such as font, font size, and spacing have been adjusted for redable. The resolution of figures has been raised to be visible clearly. Comments of referee 2: In general English is not good. Even tough I am not an English mother tongue, I have identified several errors and mistakes and the authors have to carefully check the language. Author's response: we carefully have checked all language of this paper and corrected errors and mistakes.
Please also note the supplement to this comment:
https://www.nat-hazards-earth-syst-sci-discuss.net/nhess-2018-104/nhess-2018-104-AC4-supplement.pdf

———————————————————

[Figure]

**Supplement:**

**The susceptibility assessment of multi-hazard in the Pearl River Delta Economic Zone, China**

Chuanming Ma*, Xiaoyu WU, Bin LI, Ximei Hu

*Corresponding author at: School of Environmental Studies, China University of Geosciences, Wuhan 430074, China. Tel.: +86-27-67883159. Email: machuanming@cug.edu.cn

**Abstract**

The multi-hazard susceptibility assessment can provide a basis to decision-makers for land use planning and geo-hazards management. The main scope of this paper is to assess multi-hazard susceptibility and identify susceptibility area by using an integrated method of the Analytic Hierarchy Process (AHP) and the Difference Method (MD) within MapGIS environment. The basic principle of this method is to predict future geological hazards based on occurrence mechanism and formation conditions of geological hazards and the geological conditions within the study area. Typical geo-hazards susceptibility are separately assessed by applying Analytic Hierarchy Process (AHP). The multi-hazard susceptibility is completed by synthesizing individual geo-hazards susceptibility result with the Difference Method (MD), the multi-hazard susceptibility map is generated by utilizing MapGIS platform. The multi-hazard map can provide decision-makers with visual information for geo-hazards prevention, which reduce confusion of decision-makers on high number of individual geo-hazard maps. The study area was categorized into high susceptibility zone, moderate susceptibility zone, low susceptibility zone, and insusceptible zone, accounting for 16.5%, 41.6%, 33.8% and 8.1% of the total study area, respectively. The multi-hazad susceptibility result can be combined with other conditions to provide decision- makers with theoretical basis for geo-hazards management and planning of development.

**Key words**: susceptibility assessment; mul-hazards; Analytic Hierarchy Process (AHP) - Difference Method (DM); MapGIS; The Pearl River Delta Economic Zone

**1. Introduction**

Geological hazards occur frequently, and the types of disasters in China are various (National Disaster Mitigation Center Disaster Information Department, 2009), especially southwest region of China (Tang and Wu, 1990). The Pearl River Delta Economic Zone is the transitional belt and sensitive belt of geological environment, nears the South China Sea, characterized by strong land-ocean interaction, widely distributed Quaternary, complex geological structure, and various landform, where it is susceptible to cause geological disasters (Li, 2012). With the rapid economic development for the Pearl River Delta Economic Zone, the strength of development and utilization for geological environment trends to increase, the frequency and intensity of geological hazards intensifies rapidly, which has a great threaten upon people's lives and property (Zhang, 2012). The occurrence of geological hazards seriously restricted the urban development and the sustainable development of human society (Unitto and Shaw, 2016). According to geological survey result, geo-hazards occurred in the Pearl River Delta Economic Zone mainly include collapse, landslide, debris flow, ground subsidence, karst collapse, water and soil erosion and seawater intrusion, the scale of debris flow is small, so it is not considered as object of study in this paper. Therefore, in order to minimize the economic loss and reduce threaten on people's lives and property, management of geological hazards is essential. Thus, it is very meaningful to evaluate geological hazards susceptibility and identify different susceptibility areas for prevention and management of geological hazards.

Since geological hazards are a complex phenomena, currently, various researches have focused on a single geological hazard research (Komac, 2006; Pradhan et al., 2016; Wang et al., 2015; Zhou et al., 2002). But, one region may suffer from more than one geological hazard. The multi-hazard susceptibility assessment that consists of relation between different hazards is an important tool for geological hazards management and urban planning. The United Nations (UN, 2002) has emphasized the significance of multi-hazard assessment and referred that "it is an essential element of a safer world in the twenty-first century". However, multi-hazard susceptibility assessment is a complex process and confronts with challenges. At early stages, qualitative assessment methods were widely used to evaluate geological hazards susceptibility (Bijukchhen et al., 2013; Cui et al., 2004; Degg, 1992; Liang et al., 2011;

Zhou et al. 2002), which are based on statistical analysis for the scale and density of occurred geological hazards, but it is difficult to describe the affect of different influencing factors on geological hazards. In recent years, with development of science and technology, the methods that combines qualitative and quantitative analysis are widely used to evaluate geological hazards susceptibility (Lee et al., 2018; Wang et al., 2015; Yilmaz, 2009). One widely used method for susceptibility assessment is the Analytic Hierarchy Process (AHP) ( Karaman, 2015; Karaman and Erden, 2014; Komac, 2006; Peng et al., 2012; Rozos et al., 2011). The AHP is a multiple criteria decision-making that combines qualitative and quantitative factors for ranking and evaluating alternative scenarios, among which the best solution is ultimately chosen (Satty, 1980; Satty, 2008). Preventive measures for different geological hazards are various, and their damage on environment and people's lives and property is not neutralized. Thus, multi-hazard susceptibility assessment is completed by synthesizing all individual geological hazards susceptibility result with the Difference Method. The major principle of the Difference Method is that the multi-hazard susceptibility in this a unit is considered high, as long as there is a kind of geological hazard are under high susceptibility in specific evaluation unit.

In this paper, a new method that integrated the Analytic Hierarchy Process (AHP) and the Difference Method is proposed to assess multi-hazard susceptibility. Individual hazard susceptibility is assessed with via of the Analytic Hierarchy Process (AHP) and spatial analysis of MapGIS, based on the geological hazards investigation and geological environmental conditions of the study area. The Difference Method is used to assess multi-hazard susceptibility by synthesizing the five aforementioned geohazard susceptibility assessments. Moreover, a multi-hazard susceptibility map is produced with MapGIS. The multi-hazard susceptibility map will benefit local governments in making policies on urban development and infrastructure layout, and it also offer more accurate and effective theoretical guide to land use planning and site selection of major projects, coming true the maximum utilization of limited resources and the maximum economic efficiency with limited environment.

**2. The study area**

**2.1 Natural geographical conditions**

The Pearl River Delta Economic Zone, with a total area of 41698 km$^2$, is located in the south-central Guangdong Province, China (Fig.01), nears the South China Sea, between 21°43' ~ 23°56' N latitude and 112°00' ~ 115°24' E longitude. It includes 9

prefecture-level cities.

[Figure]

Fig.01 The spatial distribution map of geo-hazards in the study area

The study area belongs to subtropical monsoon climate, characterized by mild, humid and abundant rainfall. The rainfall is characterized by high precipitation, more rainy days, stronger seasonal rainfall, and uneven spatial distribution under influence of monsoon climate. The annual precipitation is reported as about 1800-2200mm (Fig.02).

[Figure]

Fig.02 The spatial distribution map of precipitation for the study area

The topography is dominated by the Pearl River delta plain, surrounded by intermittent mountain and hills, such as Gudou Mountain, Tianlu Mountain and Luofu

Mountain. The terrain is flat, ranging in altitude from -0.2 m to 0.9 m in the plain area.

Based on the different genetic type, the geomorphic units are divided into 12 kinds of level II geomorphological units, consisting of erosion and denudation middle mountains, erosion and denudation low mountains, erosion and denudation hills, erosion and denudation platforms, karst hills, volcanic hills, delta plain, alluvial and marine deposition plain, alluvial plain, alluvial and dilluvial plain, marine deposition plain and islands.

[Figure]

Fig.03 The topography map of the study area

**2.2 Geological conditions**

**2.2.1 Essential geological condition**

Development of the strata is relatively complete, and it is characterized by complicated types and the wide distribution. The stratigraphic age of the outcropped bedrock ranges from the Metamorphic rocks to the Quaternary loose debris deposition rocks, the outcropped strata is mainly Quaternary, followed by the Sinian, Cambrian, Devonian, Carboniferous, Jurassic and Cretaceous, and the distribution of Mesoproterozoic, Ordovician, Permian and Paleogene are sporadic (Fig.04). The area of outcropped Quaternary loose accounts for 3/4 of the strata area, the outcropped bedrock area accounts for 1/4 of the strata area. The Magmatic rocks is dominated by intrusive rocks, volcanic rocks only develop in small areas. The area of Magmatic rocks accounts for about 30% of the entire study area.

The study area belongs to the South China fold belts, the Northern and Central Guangdong depression belt, and the main depression is the Sanshui Depression Basin. Some large fractures develop in the study area (Fig.04). These large fractures are characterized by multiple phases of activity, especially since the late Tertiary, which

 have affect on formation and evolution of the Pearl River Delta, structure development and crustal stability.

[Figure]

Fig.04 The geological map of the study area

**2.2.2 Hydrogeological conditions**

In the study area, groundwater is divided into three types: loose rock pore water, carbonate karst water and bedrock fissure water, hydrogeological characteristics are shown in Fig.05.

[Figure]

Fig.05 The hydrogeological map of the study area

**2.2.3 Engineering geological condition**

The rock-soil body is restricted by the topography, strata, lithology, geological
structure, and it is also affected by the hydrogeological conditions, natural geological
conditions. Based on the nature, origin and structural features of the rock-soil body,
the rock-soil body is divided into three types: magmatic rocks, metamorphic rocks
and sedimentary rocks. In addition, it can be also divided into gravel soil group, sandy
soil group, clay soil group and intrusive rock residual soil group, extrusive rock
residual soil group and metamorphic rock residual soil group (Fig.06).

[Figure]

Fig.06 The engineering geological map of the study area

**2.3 The major geological hazards**

According to a field geological survey, typical geological hazards that occurred within the study area mainly consist of collapse, landslide, ground subsidence, karst collapse, water and soil erosion, and seawater intrusion. As of 2014, there are 52 large-scale collapses, 35 landslides and 5 debris flow have been found in the study area. In addition to, 129 ground subsidence hazards occurred in the study area, among of them, there are 76 ground subsidence with less than 10 cm of accumulative subsidence are found. Water and soil erosion is fragmented distributed in mountainous areas, hilly areas and tableland areas, which are characterized as karst desertification, granite and small vegetation coverage. In addition, water and soil erosion is widely distributed in Longgang District, Shenzhen City and Huadu District, Guangzhou City. According to statistics, water and soil erosion covers an area of 2300km$^2$, accounting for about 4.8% of the total land area. The seawater intrusion mainly occurred in the Pearl River Estuary area. The scope of the annual seawater intrusion spread to the inland area (Yaxi Town - Hualong Town - Humen town area), and it possibly reached Guangzhou City during the drought years. According to the research (Liu 2004), the driving forces of seawater intrusion for the study area are mainly tides and runoff, followed by saltwater tides. The distribution for geological hazards is shown in Fig.1.

**2.4 Human activity characteristics**

At present, woodland covers a area of 20348.6 km$^2$, accounting for 48.8% of the total study area, cultivated land, garden plot and construction land account for 38.0%, 6.1%

and 6.1%, respectively. Current state of land use within the study area can be shown in Fig.7, vegetation type can be shown in Fig.8. Except for the Pearl River Delta plain located in the hinterland, other lower-lying hills or platforms can be reclaimed into dry land that is suitable for planting various crops, fruit trees and economic trees. In recent years, with the rapid economic development, the land use structure has changed significantly. The area of cultivated land and garden plot are declining year by year, and the construction land rapidly expand. In the background of rapid economic and social development, the land use structure still will has a great change in the future, and "the expansion of urban construction land, the massive loss of cultivated land and garden plot" will are the main features.

[Figure]

Fig.07 The land use map of the study area

[Figure]

Fig.08 The vegetation type map of the study area

**3. Materials and methods**

**3.1 Source of Data**

Data sources were provided by Geological Survey Center, Wuhan. Database was obtained by digitizing the existing maps of the study area, collected from Geological Survey Center, Wuhan.

**3.2 Methods**

**Geological hazards causal factors**

(1) Karst collapse causal factors

Obtained research results (Su, 1998; Wang, 2001) show that the formation of karst collapse is mainly affected by degree of karst development, overburden characteristics, geological structure, and groundwater activities. Karst development is basis and prerequisite for formation of karst collapse. Overburden is material basis for formation of karst collapse and controls its formation in certain degree. Thick overburden can effectively disperse pressure of the soil body on the soil hole. Compared with the thinner overburden, the thicker overburden is less prone to karst collapse, and the scale and form of karst collapse also are closely connected with the overburden thickness. Groundwater activities is the main triggering factor of
formation of karst collapse. Geological structure can control the development of karst
and can provide a good site for soil erosion, and the spatial distribution of karst
development is also closely related to the geological structure. In general, the
stretching direction of the karst collapse area is consistent with that of the geological
structure (Fu, 2009). Based on the above analysis and geological environmental
conditions for the study area, the causal factors of karst collapse include the degree of
karst development, lithology, overburden thickness, aquifer water yield property and
the distance to the fault.

(2) Landslide and collapse causal factors

Collapse differs from landslide obviously in the form of occurrence, scale and
perniciousness, but there are also internal relations between them, which make them
have strong consistency in temporal and spatial distribution. Collapse and landslide
belong to the problem of slope rock mass instability, they often associated with each
other in the cause of formation. Landslide is closely related with collapse, and they
usually occur accompanied. Collapse and landslide occur under the same geological
tectonic setting and the same stratigraphic lithology conditions, with the same
triggering factors. So the causal factor for occurrence of collapse maintain basically
consistency with that of landslide. Thus, this paper carries out the susceptibility
assessment of collapse and landslide.

According to the statistical analysis of geological hazards, the spatial distribution
characteristics of collapse is affected by topography, geological structure, stratigraphic
lithology, climatic conditions and hydrological conditions. Moreover, there is a
positive correlation between the annual number of collapse and temporal distribution
of precipitation (Deng, 2008). Topographical conditions are the prerequisites to

[revised manuscript text omitted]

**6 Validation of the results**

Sensitivity analysis is used to assess effects of the input criteria on the model output performance and also to validate the effect of changing variable conditions or parameter values on the system (Gomez and Jones 2010). Sensitivity analysis is to determine the effective weight of each parameter and compare it with the theoretical weight for verifying the information on the effect of scaled values and weights assigned to each parameter. The effective weight, called coefficient of variation, is computed using the following Eq. (4) (Napolitano and Fabbri 1996).

$$W = \frac{P_r \cdot P_W}{V} \cdot 100 \qquad\qquad (4)$$

where W is the effective weight of the parameter P, Pr and Pw are the scaled value and the weight of the parameter P, respectively, and V is the susceptibility index of geo-hazard.

The result of individual geo-hazard susceptibility assessment (e.g. collapse and landslide, Karst collapse, ground subsidence, water and soil erosion and seawater intrusion) is verified by sensitivity analysis. The sensitivity analysis result is shown in

Table 4. Table 4 represents the effective weights with the theoretical weight for individual geo-hazard susceptibility criterion. The effective weights of each parameters are slightly different from the theoretical weight assigned to individual geo-hazard susceptibility, which validates the accuracy and rationality of the assessment method.

<Table 4>

**7 Conclusions**

[revised manuscript text omitted]

**Table 4 Sensitivity analysis of individual geo-hazard susceptibility**

| Karst collpase | | | Collapse and landslide | | | Ground subsidence | | |
|---|---|---|---|---|---|---|---|---|
| Parameters | Theoretical weight (%) | Effective weight (%) | Parameters | Theoretical weight (%) | Effective weight (%) | Parameters | Theoretical weight (%) | Effective weight (%) |
| Degree of karst development | 21.0 | 24 | Topography | 33.0 | 30.6 | The thickness of deposition | 42.5 | 30.7 |
| Overburden thickness | 32.1 | 28.8 | Lithology | 33.0 | 30.0 | Aquifer water yield property | 27.0 | 29.1 |
| Lithology | 21.0 | 23.7 | The distance to faults | 20.0 | 20.8 | The deposition age of millisol | 16.1 | 20.3 |
| Aquifer water yield property | 10.0 | 8.7 | Precipitation | 14.0 | 19.6 | The distance to the fracture | 14.4 | 17.8 |
| The distance to the fracture | 15.9 | 16.7 | | | | | | |

| Water and soil erosion | | | Seawater intrusion | | | | | |
|---|---|---|---|---|---|---|---|---|
| Parameters | Theoretical weight (%) | Effective weight (%) | Parameters | Theoretical weight (%) | Effective weight (%) | | | |
| Topography | 21.4 | 22.5 | Topography | 14.4 | 14.9 | | | |
| Vegetation type | 25.0 | 24.8 | The type of Quaternary sedimentary rock | 16.1 | 16.2 | | | |
| Soil type | 30.8 | 29.6 | Groundwater table | 27.0 | 36.1 | | | |
| Precipitation | 11.9 | 14.8 | Precipitation | 42.5 | 37.5 | | | |
| the density of of river network | 10.9 | 10.7 | | | | | | |